# LLMs Meet VLMs: Boost Open Vocabulary Object Detection with Fine-grained Descriptors

**Sheng Jin[1], Xueying Jiang[1], Jiaxing Huang[1], Lewei Lu[2], Shijian Lu[1]***

[1] S-Lab, Nanyang Technological University       [2] SenseTimeResearch

{Jiaxing.Huang, xueying003, Shijian.Lu}@ntu.edu.sg

## Abstract

Inspired by the outstanding zero-shot capability of vision language models (VLMs) in image classification tasks, open-vocabulary object detection has attracted increasing interest by distilling the broad VLM knowledge into detector training. However, most existing open-vocabulary detectors learn by aligning region embeddings with categorical labels (e.g., bicycle) only, disregarding the capability of VLMs on aligning visual embeddings with fine-grained text description of object parts (e.g., pedals and bells). This paper presents **DVDet**, a Descriptor-Enhanced Open Vocabulary Detector that introduces conditional context prompts and hierarchical textual descriptors that enable precise region-text alignment as well as open-vocabulary detection training in general. Specifically, the conditional context prompt transforms regional embeddings into image-like representations that can be directly integrated into general open vocabulary detection training. In addition, we introduce large language models as an interactive and implicit knowledge repository which enables iterative mining and refining visually oriented textual descriptors for precise region-text alignment. Extensive experiments over multiple large-scale benchmarks show that DVDet outperforms the state-of-the-art consistently by large margins.

## 1 Introduction

Vision Language Models (VLMs) (Yu et al., 2022; Yuan et al., 2021; Zhai et al., 2021; Jia et al., 2021; Radford et al., 2021; Zhou et al., 2021a; Rao et al., 2021; Huynh et al., 2021) have demonstrated unparalleled zero-shot capabilities in various image classification tasks, largely attributed to the web-scale image-text data they were trained with (Radford et al., 2021; Jia et al., 2021). As researchers naturally move to tackle the challenge of open vocabulary object detection (OVOD) (Li et al., 2021; Kamath et al., 2021; Cai et al., 2022), they are facing a grand data challenge as there does not exist similar web-scale data with box-level annotations. The much less training data in OVOD inevitably leads to clear degradation in text-image alignment, manifesting in much weaker zero-shot capabilities in most open-vocabulary object detectors. An intriguing question arises: Can we leverage the superior image-text alignment abilities of VLMs to enhance OVOD performance?

Recent studies (Du et al., 2022; Feng et al., 2022) indeed resonate with this idea, attempting to distill the knowledge from VLMs to extend the vocabulary of object detectors. For example, ViLD (Gu et al., 2021) and its subsequent work (Ma et al., 2022; Zhou et al., 2022; Lin et al., 2023) enforce detectors' embeddings to be aligned with the embeddings from CLIP image encoder or text encoder. However, VLMs still clearly outperform open-vocabulary detectors while aligning visual embeddings with text embeddings of categorical labels (e.g., 'bicycle') as illustrated in Fig 1. Upon deep examination, we found that VLMs are particularly good at aligning fine-grained descriptors of object attributes or parts (e.g., 'bell' and 'pedal') with their visual counterparts, an expertise yet harnessed in existing OVOD models. Specifically, most existing OVOD methods focus on distilling coarse and category-level alignment knowledge on visual and textual embedding. They largely neglect the fine-grained and descriptor-level alignment knowledge that VLMs possess, leading to the under-utilization of VLM knowledge in the trained OVOD models.

---

*Corresponding author.

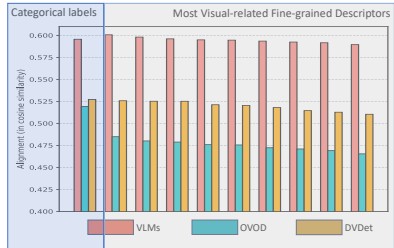 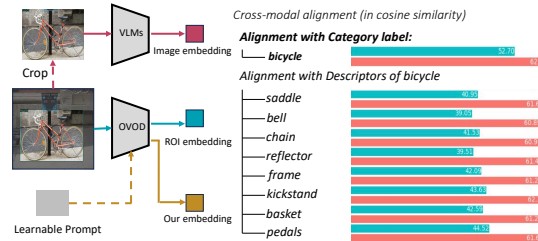

(a) Visual-textual alignment on the whole COCO dataset

(b) Illustration with an example label 'bicycle' and related textual descriptors

Figure 1: Differences in image-text alignments by VLMs and OVOD. Over the whole COCO dataset, the visual and textual embeddings from VLMs are clearly better aligned than those from OVOD (by the state-of-the-art VLDet (Lin et al., 2023)) for both categorical object labels and fine-grained descriptors as shown in (a). The proposed DVDet mines and refines fine-grained descriptors with LLMs which clearly improves region-text alignment as compared with VLDet. This can be viewed in more detail in (b) with an exemplar label 'bicycle' and fine-grained descriptors of bicycle parts. The alignment is measured by the cosine similarity between visual and textual embeddings.

We design **D**escriptor-Enhanced Open **V**ocabulary **Det**ection (DVDet) that exploits VLMs' prowess in descriptor-level region-text alignment for open vocabulary object detection. The essential idea is to exploit VLMs' alignment ability via customized visual prompt, which mines regional fine-grained descriptors from large language models (LLMs) iteratively and enables prompt training without resource-intensive grounding annotations. The key design in DVDet is **Co**nditional **Co**ntext visual **P**rompt (CCP) that transforms region embeddings into image-like counterparts by fusing contextual background information around region proposals. This allows CCP to be seamlessly integrated into various open-vocabulary detection training with little extra designs.

To train CCP effectively, we introduce LLMs as communicable and implicit knowledge repositories for iterative generation of fine-grained descriptors for precise region-text alignment. Specifically, we design a hierarchical update mechanism that interacts with LLMs by retaining most-related descriptors while actively soliciting new descriptors from LLMs, enabling CCP training without additional annotation costs. This mechanism enhances descriptor diversity and descriptor availability, leading to regional fine-grained descriptors that are tailored to the most relevant object categories. In addition, we design a simple yet effective descriptor merging and selection strategy to tackle two challenges in CCP training: 1) distinct object categories could share similar fine-grained descriptors which leads to potential confusion in CCP training; 2) object images may not have all fine-grained descriptors present due to occlusions, etc., more detail to be described in the ensuing Method.

The contributions of this work can be summarized in three major aspects. *First*, we introduce a feature-level visual prompt that transforms object embeddings into image-like representations that can be seamlessly plugged into existing open vocabulary detectors in general. *Second*, we design a novel hierarchical update mechanism that enables effective descriptor merging and selection and dynamical refinement of region-text alignment via iterative interaction with LLMs. *Third*, extensive experiments demonstrate that the proposed technique improves open-vocabulary detection substantially for both base and novel categories.

## 2 RELATED WORK

**Open-Vocabulary Object Detection (OVOD)**: OVOD utilizing the knowledge of pretrained VLMs (Radford et al., 2021) has attracted increasing attention (Zhong et al., 2022; Minderer et al., 2022) with the advance in open vocabulary image classification (Yu et al., 2022; Yuan et al., 2021; Zhai et al., 2021; Jia et al., 2021; Radford et al., 2021). For example, ViLD (Gu et al., 2021) distills knowledge from VLMs into a two-stage detector, harmonizing the detector's visual embeddings with those from the CLIP image encoder. HierKD (Ma et al., 2022) focuses on hierarchical global-local distillation and RKD (Bangalath et al., 2022) explores region-based knowledge distillation to improve the alignment between region-level and image-level embeddings. In addition, VLDet (Lin et al., 2023) and Detic (Zhou et al., 2022) align their detector embeddings with those from CLIP text encoder. Nevertheless, all these prior studies share similar misalignment between their trained detectors and pretrained VLMs: VLMs capture more comprehensive knowledge including fine-grained knowledge about object parts, object attributes, and contextual background while open vocabulary

detectors focus on learning precise localization of interested objects. Such misalignment tends to restrict the efficacy of knowledge distillation from VLMs to OVOD.

Recently, prompt-based methods such as DetPro (Du et al., 2022) and PromptDet (Feng et al., 2022) have emerged as an alternative for alleviating the misalignment between upstream classification knowledge in VLMs and downstream knowledge in detection tasks. These methods adjust the textual embedding space of VLMs to align with regional visual object features by incorporating continuous prompts within VLMs. Despite their success, the adjustment focuses on cross-modal alignment between categorical labels and ROI embedding only, which tends to disrupt the inherent visual-textual alignment properties of VLMs. We design a feature-level prompt learning technique that formulates the ROI embeddings of detectors to be highly similar to image-level embeddings, preserving the image-text alignment capabilities of pretrained VLMs effectively.

**Visual Prompt Learning** The concept of 'prompting' originates from the field of Natural Language Processing (NLP) and has gradually gained increasing attention as a means of guiding language models via task instructions (Brown et al., 2020). The evolution of continuous prompt vectors in few-shot scenarios (Li & Liang, 2021; Liu et al., 2021a) demonstrates its cross-domain applicability. Pertinently, VPT (Jia et al., 2022) and its successors (Bahng et al., 2022a;b) have extended this idea to the visual domain, achieving precise pixel-level predictions. In addition, the prompting idea has also been explored in pre-trained video recognition models as well (Ju et al., 2022; Lin et al., 2022b). However, existing visual prompt algorithms generate a single prompt for each downstream task, which cannot handle detection tasks well that often involve multiple objects in a single image. We address this issue by designing CCP which generates a conditional prompt for each object.

**Leveraging Language Language Models** Linguistic data has been increasingly exploited in open-vocabulary related research, and the recent LLMs have demonstrated their comprehensive knowledge that can be beneficial in various NLP tasks. This trend has extended to computer vision research, and several studies have been reported to investigate how LLMs can assist in downstream computer vision tasks. For example, (Menon & Vondrick, 2023; Zhang et al., 2023) have harnessed linguistic knowledge in pretrained LLMs to generate descriptors for each visual category. Such augmentation enriches VLMs without additional training or labeling efforts. Inspired by CuPL (Menon & Vondrick, 2023), CaF employs GPT-3 (Brown et al., 2020) to craft semantically enriched texts, thereby enhancing the alignment between CLIP's text and images. However, most existing research treats LLMs as a static database, acquiring useful information through a one-time interaction. We introduce a simple yet effective hierarchical mechanism that continuously interacts with LLMs during the model's training process, obtaining more diverse and visual-oriented textual data.

## 3 METHOD

### 3.1 OVERVIEW

**Problem setup.** Open Vocabulary Object Detection (OVOD) (Zareian et al., 2021) leverages a dataset of image-text pairs to broaden its detection vocabulary from pre-defined categories to novel categories. Formally, the task is to construct an object detector using a detection dataset defined as $\mathcal{T} = (\{(I_i, g_i, D_i)\}_{i=1}^{N})$, where $I_i$ represents an image, $g_i = (b_i, c_i)$ denotes the ground truth annotations consisting of bounding box coordinates $b_i$ and associated base categories $c_i \in C^{\text{base}}$, and $D_i$ symbolizes fine-grained descriptors that are generated through LLMs. The primary goal is to facilitate the detection of new classes $C^{\text{novel}}$ in the inference stage.

The predominant OVOD framework typically utilizes a two-stage detection architecture as its backbone, incorporating text embeddings to reformulate the classification layer. Generally, a popular two-stage object detector, such as Mask-RCNN, comprises a visual backbone encoder denoted as $\Phi_{\text{ENC}}$, a class-agnostic region proposal network (RPN) represented by $\Phi_{\text{RPN}}$, and an open vocabulary classification module labeled as $\Phi_{\text{CLS}}$. The overall detection can be formulated by:

$$\{\hat{y}_1, \ldots, \hat{y}_n\} = \Phi_{\text{CLS}} \circ \Phi_{\text{RPN}} \circ \Phi_{\text{ENC}}(I_i) \tag{1}$$

where $I_i$ denotes the $i$-th input image and $\{\hat{y}_1, \ldots, \hat{y}_n\}$ represents the set of predicted outputs.

Fig. 2 shows an overview of the proposed DVDet framework including a VLM-guided conditional context prompting flow and a LLMs-assisted descriptor generation flow. In the prompting flow, the network takes an input image $I_0$ and extracts the features $r_0^i$ for the $i$-th proposal. It then enlarges

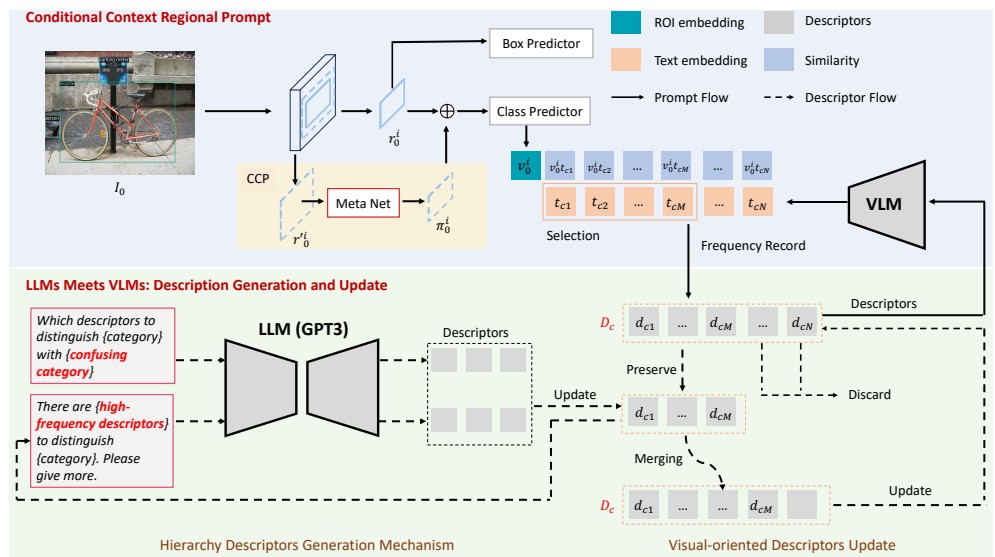

Figure 2: **Overview of our proposed DVDet framework**: DVDet comprises two specific flows to improve the region-text alignment in open vocabulary detection. In the prompt flow (denoted by the solid line), the proposed conditional context prompt (CCP) transforms the ROI embeddings into image-like representation by fusing the contextual background information around the region proposal, that can be incorporated to facilitate the training of open vocabulary detectors. In the descriptor flow (denoted by the dashed line), a hierarchy mechanism is designed to generate and update fine-grained descriptors via iterative interaction with LLMs for precise region-text alignment.

the proposal to integrate the contextual background and further extracts features to formulate $r_0^{'i}$. We design a learnable *meta-net* that takes $r_0^{'i}$ to create a region prompt $\pi_o^i$ and combines the prompt with $r_0^i$ to obtain a prompted features $v_0^i$. Finally, $r_0^i$ is fed to a *Box Predictor*, and $v_0^i$ to a *Class Predictor* to assimilate the text embeddings of category labels and their fine-grained descriptors $D_c$. In the descriptor flow, we design a hierarchy mechanism to generate and update $D_c$ for each category via iterative interactions with LLMs in training. Specifically, $D_c$ records the frequency of fine-grained descriptors as well as their probability of being misclassified to other categories. It also records confusing categories that statistically have high misclassification probability. During the training process, $D_c$ retains high-frequency fine-grained descriptors while discarding low-frequency ones. It employs both high-frequency descriptors and confusing categories to prompt LLMs to generate more diverse and visually relevant descriptors, which are further incorporated into $D_c$ with a semantic merging process.

## 3.2 Conditional Context Regional Prompts

In this section, we introduce the **C**onditional **C**ontext Regional **P**rompts method (CCP), a strategy designed to bridge the gap between pretrained foundational classification models and downstream detection tasks. This technique uses the surrounding contextual background information to transform ROI features into image-like features. Importantly, since current detectors excel at finding unfamiliar objects but have difficulty classifying them accurately, the CCP is integrated only into the classification branch of existing detectors. This improves their accuracy without affecting the localization branch's ability to identify as numerous unknown targets as possible.

Given a pre-trained backbone $E$ and a dataset for downstream tasks, we extract features for an image $I_0$ as $R_0 = [r_0^1, r_0^2, \cdots, r_0^M]$ where $r_0^i = E(I_0(b_0^i))$. Notably, $M$ represents the number of region proposals and $b_0^i$ signifies the $i$-th proposal. Our objective is to develop a region-conditional visual prompt $\pi_0^i$ for the $i$-th detected proposal, more details to be elaborated in the ensuing subsection.

**Prompt Design.** In classification tasks, the visual prompt mechanism (Jia et al., 2022) learns a dataset-specific prompt for each task. But for detection tasks, we require a mechanism that can create a contextual conditional prompt $\pi_0^i$ for $i$-th detected proposal. Considering the varying scales and quantities of proposals across different samples, we adopt convolutional layers to build a lightweight

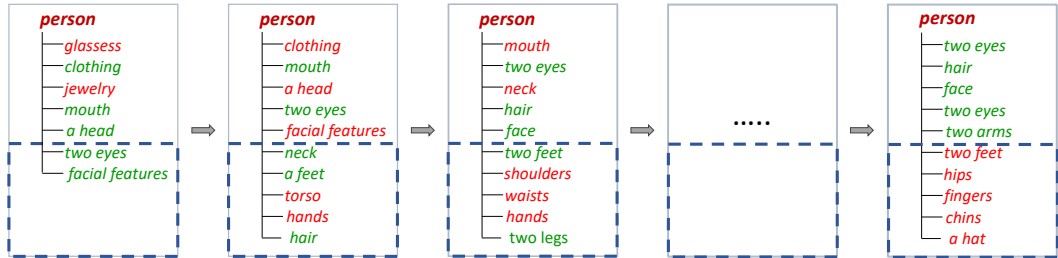

Figure 3: The iterative update of fine-grained descriptors. In the training stage, we continuously generate new fine-grained descriptors (highlighted in blue boxes) via interaction with LLMs. With the recorded usage frequency, high-frequency descriptors (highlighted in green) are preserved and low-frequency descriptors (highlighted in red) are discarded. We can observe that certain fine-grained descriptors such as 'hair', 'two eyes', and 'face' are consistently preserved after generation while visually irrelevant descriptors such as 'jewelry' are only generated at early stage and then discarded.

*meta-network*, that is adept at processing a variety of object proposals. For each proposal $b_0^i = (x_1, x_2, y_1, y_2)$, we merge the surrounding background information where the background region is defined by $b_0^{'i} = (x_1', x_2', y_1', y_2')$, calculated as follows:

$$x_1' = x_1 - m, y_1' = y_1 - n$$
$$x_2' = x_2 + m, y_2' = y_2 + n \tag{2}$$

where $m$ and $n$ are constants. Next, we extract the features $r_0^{'i}$ from the expanded region, and the *meta-network* learns the regional visual prompt $\pi_0^i$ using the formula $\pi_0^{'i} = h_\theta(r_0^{'i})$, and $h_\theta(.)$ represents the *Meta-Net* parameterized by $\theta$. Finally, the learned prompt is combined with the feature $r_0^i$ to create a more detailed prompted feature $v_0^i = r_0^i + \pi_0^{'i}$.

## 3.3 LLMS MEETS VLMS

In this section, we treat the LLMs as interactive implicit knowledge repositories to generate fine-grained descriptors for CCP training. Specifically, we design a hierarchical generation mechanism that interacts with LLMs iteratively to generate more diverse and visually relevant category descriptions throughout the training process, with more detail to be elaborated in the ensuing subsections.

**Descriptors Initialization.** We adopt a similar input protocol as (Menon & Vondrick, 2023) to prompt LLMs. For each category denoted as $c$, we extract its fine-grained descriptors $D_c$, represented as $D_c = [d_{c1}, d_{c2}, \cdots, d_{cK}]$, accompanied by the corresponding text embeddings $T_c = [t_{c1}, t_{c2}, \cdots, t_{cK}]$, where $K$ signifies the quantity of fine-grained descriptors. For all categories, we obtain a fine-grained descriptor dictionary $D = [D_1, D_2, ......, D_M]$.

**Descriptors Record.** The fine-grained descriptors are used for the category prediction. Since we cannot guarantee the presence of each descriptor in every sample, we introduce a semantic selection strategy for each proposal. The selection function $s(c, I_o^i)$ is defined by:

$$s(c, I_o^i) = \frac{1}{N} \sum_{d \in Rank_N(D_c)} \phi(d, I_o^i) \tag{3}$$

where, $\phi(d, I_o^i)$ represents the probability of how the descriptor $d$ is relevant to the $i$-th proposal of image $I_0$, and $Rank_N$ selects the top $N$ descriptors based on the value of $\phi(d, I_o^i)$. For the $i$-th proposal, we predict its category label via $\arg\max_{c \in C} s(c, I_o^i)$. For each category $c$, we record the usage frequency of each descriptor and its probability of being misclassified to other categories (i.e., the confusing categories with high misclassification probability).

**Descriptors Hierarchy Generation and Update.** During the training stage, we generate fine-grained descriptors via a hierarchy mechanism at intervals of every $N$ iterations. The updating of fine-grained descriptors consists of two core operations. First, we record the usage frequency of the descriptor according to Eq. 3. The high-frequency descriptors are preserved and the low-frequency descriptors are discarded. Second, we prompt LLMs with an input template that consists of high-frequency descriptors to gather descriptors designed as follows:

```
Q: There are several useful visual features to tell there is a
   {category name} in a photo, including {the first frequency
   descriptors, the second frequency descriptors, ...}.
```

where {category name} is substituted for a given category label $c$. The generated list then constitutes the descriptor dictionary $D$. Subsequently, we further generate fine-grained descriptors for this category to differentiate it from confusing classes. We prompt LLMs with an input template that consists of the confusing categories:

```
Q: Which visual features could be used to distinguish
   {category name} from some confusing categories including
   {confusing category 1, confusing category 2, confusing
   category 3, ...} in a photo?
```

The newly generated descriptors further expand the descriptor dictionary $D$. However, some newly generated descriptors $d_i$ may already exist in $D$ already. Further, including it into $D$ may lead to the presence of the same descriptor in multiple categories, leading to potential semantic confusion during training. We address this issue by measuring the cosine similarity $s_{ij}$ between $d_i$ and $D$. If $s_{ij} > \gamma$, we merge the descriptor's text embedding via $t_j = \alpha t_i + (1 - \alpha)t_j$, where $\gamma$ is a constant and $\alpha$ is the momentum coefficient.

## 4 EXPERIMENTS

### 4.1 DATASETS

We evaluated DVDet over two widely adopted benchmarks, *ie*, COCO (Lin et al., 2014) and LVIS (Gupta et al., 2019). For the COCO dataset, we follow OV-RCNN (Zareian et al., 2021) to split the object categories into 48 base categories and 17 novel categories. As in (Zareian et al., 2021), we keep 107,761 images with base class annotations as the training set and 4,836 images with base and novel class annotations as the validation set. Following (Gu et al., 2021; Zareian et al., 2021), we report mean Average Precision (mAP) at an IoU of 0.5. For the LVIS dataset, we follow ViLD (Gu et al., 2021) to split the 337 rare categories into novel categories and the rest common and frequent categories into base categories (866 categories). Following (Lin et al., 2023), we report the mask AP for all categories. For brevity, we denote the open-vocabulary benchmarks based on COCO and LVIS as OV-COCO and OV-LVIS.

### 4.2 IMPLEMENTATION DETAILS

In our experiments, we employ pre-trained models in prior studies as the base and include our prompt learning techniques on top of them for evaluations. Specifically, we employ the CLIP text encoder to encode both categorical labels and their fine-grained descriptors. *In the open-vocabulary COCO experiments*, we follow the OVR-CNN setting (Zareian et al., 2021) without any data augmentation and adopt Faster R-CNN (Ren et al., 2015) with ResNet50-C4 (He et al., 2016) as the backbone. For the warmup, we increase the learning rate from 0 to 0.002 for the first 1000 iterations. The model is trained for 5,000 iterations using SGD optimizer with batch size 8 and the learning rate is scaled down by a factor of 10 at 6000 and 8000 iterations. *In open-vocabulary LVIS experiments*, we follow Detic (Zhou et al., 2022) to adopt CenterNet2 (Zhou et al., 2021b) with ResNet50 (He et al., 2016) as backbone. We use large-scale jittering (Ghiasi et al., 2021) and repeat factor sampling as data augmentation. For the warmup, we increase the learning rate from 0 to 2e-4 for the first 1000 iterations. The model is trained for 10,000 iterations using Adam optimizer with batch size 8. All expriments are conducted on 4 NVIDIA V100 GPUs. More details can be found in Appendix.

### 4.3 OPEN-VOCABULARY DETECTION ON COCO

Table 1 shows the performance of different methods on the open-vocabulary COCO datasets. It can be seen that our method, when incorporated into multiple existing open-vocabulary detectors, achieves stable performance improvements consistently. This indicates that introducing alignment with fine-grained descriptors can effectively enhance the performance of existing open-vocabulary detectors. The baseline method utilizes the pretrained RPN (Zhong et al., 2022) to extract proposals and directly feed them into CLIP for classification. We can observe that CLIP achieves good accuracy on novel classes, reaffirming its powerful zero-shot capabilities. As a comparison, state-of-the-art OVOD methods experience sharp accuracy drops while handling objects of novel classes,

Table 1: Open-vocabulary object detection on COCO dataset. Including our DVDet improves the state-of-the-art consistently for both base and novel classes. 'Baseline' utilizes the pretrained RPN (Zhong et al., 2022) to extract proposals and directly feeds them into CLIP for classification. 'Novel AP' indicates the zero-shot performance.

| Method | Novel AP | Base AP | Overall AP |
|---|---|---|---|
| Baseline | 29.7 | 24.0 | 25.5 |
| ViLD (Gu et al., 2021) | 27.6 | 59.5 | 51.3 |
| +DVDet | 29.3 | 60.6 | 52.4 |
| Detic (Zhou et al., 2022) | 27.8 | 51.1 | 44.9 |
| +DVDet | 29.5 | 53.6 | 47.3 |
| RegionCLIP (Zhong et al., 2022) | 26.8 | 54.8 | 47.5 |
| +DVDet | 28.4 | 56.6 | 49.2 |
| VLDet (Lin et al., 2023) | 32.0 | 50.6 | 45.8 |
| +DVDet | 34.6 | 52.8 | 48.0 |
| BARON (Wu et al., 2023) | 33.1 | 54.8 | 49.1 |
| +DVDet | 35.8 | 57.0 | 51.5 |

Table 2: Open-vocabulary object detection on LVIS dataset using ResNet50 (RN50) (He et al., 2016) and Swin-B (Liu et al., 2021b) as backbones. 'Baseline' utilizes the pretrained RPN (Zhong et al., 2022) to extract proposals and directly feeds them into CLIP for classification. $mAP_{Novel}^{mask}$ indicates the zero-shot performance.

| Method | Backbone | $mAP_{Novel}^{mask}$ | $mAP_{c}^{mask}$ | $mAP_{f}^{mask}$ | $mAP_{all}^{mask}$ |
|---|---|---|---|---|---|
| Baseline | RN50 | 11.6 | 9.6 | 7.6 | 9.2 |
| ViLD (Gu et al., 2021) | RN50 | 16.6 | 24.6 | 30.3 | 25.5 |
| +DVDet | RN50 | 18.7 | 25.8 | 31.6 | 27.1 |
| DetPro (Du et al., 2022) | RN50 | 19.8 | 25.6 | 28.9 | 25.9 |
| +DVDet | RN50 | 21.3 | 28.2 | 31.3 | 28.1 |
| RegionCLIP (Zhong et al., 2022) | RN50 | 17.1 | 27.4 | 34.0 | 28.2 |
| +DVDet | RN50 | 19.1 | 29.2 | 35.2 | 29.6 |
| VLDet (Lin et al., 2023) | RN50 | 21.7 | 29.8 | 34.3 | 30.1 |
| +DVDet | RN50 | 23.1 | 31.2 | 35.4 | 31.2 |
| BARON (Wu et al., 2023) | RN50 | 19.2 | 26.8 | 29.4 | 26.5 |
| +DVDet | RN50 | 21.3 | 28.7 | 31.8 | 28.3 |
| Detic (Zhou et al., 2022) | Swin-B | 23.9 | 40.2 | 42.8 | 38.4 |
| +DVDet | Swin-B | 25.2 | 41.4 | 44.6 | 40.4 |
| VLDet (Lin et al., 2023) | Swin-B | 26.3 | 39.4 | 41.9 | 38.1 |
| +DVDet | Swin-B | 27.5 | 41.8 | 43.2 | 40.2 |

and this applies to various existing OVOD approaches that seek broader cross-modal alignment with dataset caption annotations (Lin et al., 2023), introduction of classification datasets (Zhou et al., 2022) and construction of a concept pool (Zhong et al., 2022). The accuracy drop clearly highlights the necessity of optimizing the classifier in open vocabulary object detectors. By introducing the alignment with fine-grained textual descriptions of categories, our method serves as a general plugin that can complement existing open-vocabulary object detectors consistently and effectively.

## 4.4 OPEN-VOCABULARY DETECTION ON LVIS

Table 2 shows open-vocabulary detection on LVIS dataset. Similar to the experiments on COCO dataset, the "Base-only" still performs better on the novel classes. However, its performance generally falls below state-of-the-art OVOD methods due to the higher complexity of object detection tasks. Nevertheless, our proposed method complements existing methods consistently, for both ViLD-type methods and prompt-based methods such as DetPro (Du et al., 2022). This shows that compared to the current strategies involving prompts in text encoders, our approach serves as an effective complement. We further examine the generalization of our method by adopting Swin-B as the backbone. Experiments show that incorporating our method into Detic and VLDet improves the accuracy by around 1.3% on novel categories consistently.

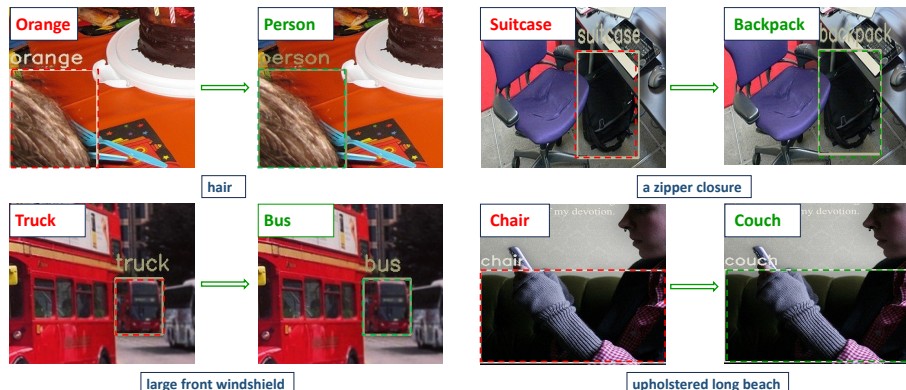

Figure 4: Introducing our fine-grained text descriptors (shown at the bottom of each sample) improves the open-vocabulary detection consistently especially under challenging scenarios with distant or occluded objects, small inter-class variations, etc. For each of the four sample images, the red-color class names at the top-left corner of the first image are predictions without our method, and the green-color class names in the second image are predictions after including our method. The red/green boxes within the sample images show related detection. Close-up view for details.

Table 3: Ablation studies of our designs in DVDet on the OV-COCO benchmark with VLDet and RegionCLIP as two base networks. We use two input 'Templates' to interact with LLMs to obtain fine-grained descriptors. 'Template H' includes high-frequency descriptors, and 'Template C' includes confusing categories, 'Prompt' denotes the proposed conditional context region prompt.

| Fine-grained Descriptors | | Prompt | VLDet | | RegionCLIP | |
|---|---|---|---|---|---|---|
| Template H | Templete C | | $mAP_{novel}$ | $mAP_{base}$ | $mAP_{novel}$ | $mAP_{base}$ |
| | | | 32.0 | 50.6 | 26.8 | 54.8 |
| ✓ | ✓ | | 29.7 | 48.7 | 24.6 | 51.0 |
| | | ✓ | 33.1 | 51.2 | 27.3 | 55.2 |
| | ✓ | ✓ | 34.0 | 52.2 | 28.0 | 56.2 |
| ✓ | | ✓ | 33.8 | 52.3 | 27.6 | 55.9 |
| ✓ | ✓ | ✓ | 34.6 | 52.8 | 28.4 | 56.6 |

**Visualization.** We show how introducing fine-grained descriptors improves the open-vocabulary detection qualitatively. As Fig. 4 shows, including our design improves the detection significantly while facing challenging scenarios with distant or occluded objects, small inter-class variations, etc. With fine-grained descriptors such as hair, zippers, and large glass front windshield, our model can better align with the text space and enable more accurate recognition and understanding while handling novel classes. In Fig. 5, we further show how our model progressively aligns targets with relevant descriptors (e.g., the word 'school bus' on the vehicle) of new categories (along the training process), thereby reducing ambiguity and misclassifications (airplane → car → bus) effectively. More visualization results can be found in Appendix.

## 4.5 ABALTION STUDIES

In this section, we conduct ablation studies on OV-COCO benchmark using VLDet (Lin et al., 2022a) and RegionCLIP (Zhong et al., 2022) as two base networks, respectively.

**Component Analysis.** We examine the effectiveness of different components in DVDet on the OV-COCO benchmark. As Table 3 shows, the contribution of fine-grained categorical descriptors is compromised clearly at the absence of prompt learning (i.e., *Prompt*), and this is well aligned with the statistical data in Fig. 1. In addition, incorporating the prompting with fine-grained descriptors improves the detection performance significantly, substantiating the benefits of fine-grained descriptors to visual-textual alignment. Besides, we adopt two templates to interact with LLM to generate fine-grained descriptors. The template using the confusing categories outperforms that using high-frequency descriptors slightly, largely due to the synergy of the semantic selection with the frequency-based filtering mechanism that plays a critical role in filtering out irrelevant descriptors and partially ensures the reliability of the overall descriptor pool in training. Further, DVDet outperforms other variants consistently, demonstrating the synergy of merging fine-grained descriptors generated from different templates with prompt learning which enriches the training with a more comprehensive understanding of object categories.

Table 5: Ablation study on interaction strategies with Knowledge Base (LLMs).

| Interaction Strategy | VLDet | | RegionCLIP | |
|---|---|---|---|---|
| | $\text{mAP}_{novel}$ | $\text{mAP}_{base}$ | $\text{mAP}_{novel}^{mask}$ | $\text{mAP}_{base}^{mask}$ |
| Static Knowledge Base | 33.2 | 51.9 | 27.6 | 55.1 |
| Interactive Knowledge Base | 34.6 | 52.8 | 28.4 | 56.6 |

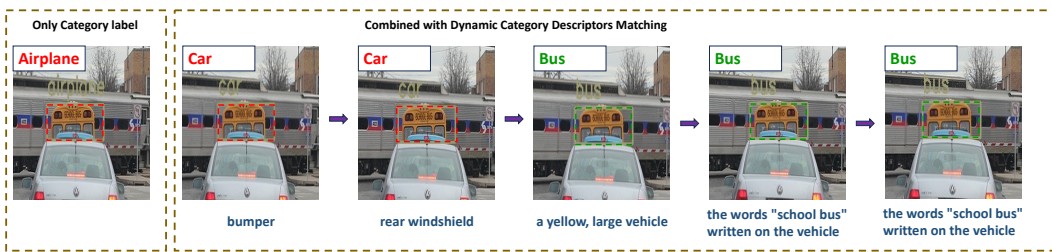

Figure 5: Object detection is improved progressively with iterative extraction of fine-grained descriptors from LLMs and matching them with the detected target. Texts at the top-left corner of each sample show recognized classes, and texts at the bottom show extracted fine-grained descriptors.

**Transfer To Other Datasets.** To ascertain the generalization of the proposed DVDet, we apply our COCO-trained model to the test set of PASCAL VOC (Everingham et al., 2010) and the validation set of LVIS with little additional training. This is achieved by utilizing the context conditional prompts from the OV-COCO and modifying the class embeddings of the classifier head accordingly. PASCAL VOC contains 20 object categories including 9 absent in COCO, thereby presenting a notable challenge while transferring models without aids from any supplementary training images, not to mention the inherent domain gap. LVIS dataset boasts a substantial catalogue of 1203 object categories, vastly exceeding the label space in COCO. Despite these challenges, DVDet demonstrates remarkable effectiveness across diverse image domains and language vocabularies, as evidenced by 2.3% and 2.1% improvements on the two new datasets as shown in Table 4. It should be highlighted that though many LVIS category names are absent in COCO, DVDet succeeds in learning close descriptors and thereby facilitating a smoother transition while adapting to the LVIS benchmark.

Table 4: Transfer to other datasets. We evaluated COCO-trained model on PASCAL VOC (Everingham et al., 2010) test set and LVIS validation set without re-training. We report mAP at an IoU of 0.5.

| Method | PASCAL VOC | LVIS |
|---|---|---|
| VLDet | 61.7 | 10.0 |
| +DVDet | 64.0 | 12.1 |
| RegionCLIP | 46.9 | 6.1 |
| +DVDet | 48.2 | 7.8 |

**Effectiveness of Interactive Knowledge Base.** The successful creation of fine-grained descriptors depends on iterative interaction with LLMs in training. We validate this by comparing it with a one-time interaction, where LLMs act like a static knowledge base. Specifically, we employ confident samples from CAF (Menon & Vondrick, 2023) to obtain visual-related fine-grained descriptors. As Table 5 shows, the iterative interaction (34.6 AP and 28.4 AP for novel classes) outperforms the static method (33.2 AP and 27.6 AP) clearly. This reconfirms that the dynamic interaction allows LLMs to better understand the detector's requirements, offering more trustworthy descriptors.

## 5 CONCLUSIONS

This paper presents DVDet, an innovative open vocabulary detection approach that introduces fine-grained descriptor for better region-text alignment and open-vocabulary detection. DVDet consists of two key designs. The first is Conditional Context regional Prompt (CCP), which ingeniously transforms region embeddings into image-like representations by merging contextual background information, enabling CCP to be seamlessly integrated into open vocabulary detection with little extra designs. The second is a hierarchical descriptor generation that iteratively interacts with LLMs to mine and refine fine-grained descriptors according to their performance in prompt training. Without any resource-intensive grounding annotations, DVDet coordinates LLMs-assisted descriptor generation and VLM-guided prompt training effectively. Extensive experiments show that DVDet improves the performance of existing open vocabulary detectors consistently. Moving forwards, we plan to investigate the synergy between powerful foundational models including LLMs and VLMs, for various open vocabulary dense prediction tasks.

## 6 ACKNOWLEDGEMENT

This study is supported under the RIE2020 Industry Alignment Fund – Industry Collaboration Projects (IAF-ICP) Funding Initiative, as well as cash and in-kind contributions from the industry partner(s).

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
