# Supplementary Material: LLMs Meet VLMs: Boost Open Vocabulary Object Detection with Fine-grained Descriptors

**Sheng Jin**[1], **Xueying Jiang**[1], **Jiaxing Huang**[1], **Lewei Lu**[2], **Shijian Lu**[1*]
[1] S-Lab, Nanyang Technological University      [2] SenseTimeResearch
{Jiaxing.Huang, xueying003, Shijian.Lu}@ntu.edu.sg

## A  Appendix

### A.1  Implementation Details

#### A.1.1  Parameter Settings

**Conditional Context Prompt.** In our experiments, we employ two constants, $m$ and $n$, to determine the size of the enlarged bounding boxes. For the $i$-th detected proposal, the background region is defined as $b_0^{'i} = (x_1', x_2', y_1', y_2')$, computed by:

$$
\begin{aligned}
x_1' &= x_1 - m, \quad y_1' = y_1 - n, \\
x_2' &= x_2 + m, \quad y_2' = y_2 + n,
\end{aligned}
\tag{1}
$$

Both $m$ and $n$ are constants and are fixed at a value of 20. The MetaNet employs a single-layer $3 \times 3$ convolution.

**Descriptors Update and Generation.** We update the fine-grained descriptors at intervals of every 200 iterations. Specifically, for each detected object, we select the top 5 high-frequency descriptors of each category and record their usage upon selection. Only the top 5 most frequent descriptors from prior training phases are retained during each update, while others are discarded. In the semantic merging phase, the similarity threshold $\gamma$ remains set at 0.95, and we adopt a momentum coefficient of 0.5 without the need for extensive parameter tuning.

#### A.1.2  Architecture Details of Box Predictor and Class Predictor

Due to constraints in the main manuscript, we provided a brief overview there. Here, we present the detailed architecture, as shown in Fig. 1.

Table 1: Ablation studies on semantic selection of DVDet on the OV-COCO benchmark using VLDet as the base networks. 'top $N$' refers to the fine-grained descriptors used for category prediction in Eq. **??**.

| top $N$ | 0 | 1 | 3 | 5 | 10 | 15 |
|---|---|---|---|---|---|---|
| $\text{mAP}_{base}$ | 50.6 | 51.2 | 52.0 | **52.8** | 51.9 | 51.7 |
| $\text{mAP}_{novel}$ | 32.0 | 32.4 | 34.4 | **34.6** | 33.0 | 32.2 |

### A.2  Component Analysis

In this section, we conduct component analysis on the OV-COCO benchmark using the VLDet method (Lin et al., 2022).

---

*Corresponding author.

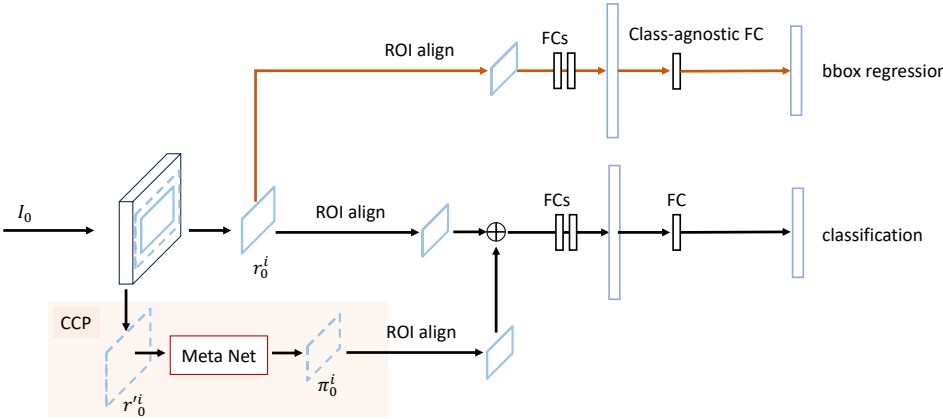

Figure 1: In this setting, we show three architectures of the proposed context conditional prompt mechanism.

Table 2: Ablation studies on frequency-based update mechanism of DVDet on the OV-COCO benchmark using VLDet as the base networks.

| the number of high-frequency descriptors | 1 | 3 | 5 | 10 | 15 |
|---|---|---|---|---|---|
| $\text{mAP}_{base}$ | 51.7 | 52.4 | **52.8** | 52.6 | 52.4 |
| $\text{mAP}_{novel}$ | 33.4 | 34.2 | **34.6** | 34.5 | 34.0 |

### A.2.1 EFFECTIVENESS OF SEMANTIC SELECTION MECHANISM.

Given that we cannot ensure the presence of each descriptor in every sample, we employ a semantic selection strategy tailored for each proposal. The selection function $s(c, I_o^i)$ is delineated in Eq. **??**. This mechanism aims to mitigate the potential adverse effects caused by fine-grained descriptors that do not pertain to the detected input object. As indicated in Table 1, the performance declines when $\text{top}N \geq 10$ due to the introduction of irrelevant descriptors.

### A.2.2 EFFECTIVENESS OF THE FREQUENCY-BASED PRESERVE MECHANISM.

The frequency-based Preserve Mechanism has two pivotal roles: (i) ensuring a sufficient number of visually related fine-grained descriptors within the descriptor pool, and (ii) embedding these descriptors into an input template to guide the LLMs toward generating more visually related descriptors. As evidenced by Table 1, while performance does improve with the count of high-frequency descriptors $N$, the enhancement becomes marginal when $N > 5$.

### A.2.3 EFFECTIVENESS OF SEMANTIC MERGING MECHANISM.

The goal of the semantic merging mechanism is to reduce the chance of the same descriptor appearing in different categories, which might cause confusion during training. To solve this, we calculate the cosine similarity $s_{ij}$ between a new descriptor $d_i$ and existing descriptors in dictionary $D$. If $s_{ij} > \gamma$, we combine the text embeddings of the descriptor using $t_j = \alpha t_i + (1 - \alpha)t_j$, where $\gamma$ is a constant and $\alpha$ is the momentum coefficient. As shown in Table 3, without the semantic merging

Table 3: Ablation studies on semantic merging of DVDet on the OV-COCO benchmark using VLDet as the base networks. $\gamma$ refers to the similarity threshold.

| $\gamma$ | 0.9 | 0.95 | 1.1 |
|---|---|---|---|
| $\text{mAP}_{base}$ | 52.3 | **52.8** | 51.3 |
| $\text{mAP}_{novel}$ | 34.0 | **34.6** | 32.8 |

Table 4: Ablation study on the proposed CCP mechanism.

| CCP | VLDet | | RegionCLIP | |
|---|---|---|---|---|
| | $mAP_{novel}$ | $mAP_{base}$ | $mAP^{mask}_{novel}$ | $mAP^{mask}_{base}$ |
| w/o Background Information | 33.4 | 51.8 | 27.8 | 55.5 |
| Background Information Fusion | 34.6 | 52.8 | 28.4 | 56.6 |

$\gamma = 1.1$, the improvement compared to VLDet (Lin et al., 2023) is not significant. The best results are achieved when $\gamma = 0.95$. Because the main purpose of semantic merging is to avoid having the same descriptor in multiple categories, we did not do extensive experiments to determine the best parameters.

### A.2.4   EFFECTIVENESS OF BACKGROUND FUSION PROMPT.

In our experiments, we expand the detected proposal to incorporate background information. Specifically, we compare our method with a version that does not integrate the background (i.e., both $m$ and $n$ in Eq. 1 are set to 0). The experimental results underscore the importance of incorporating background information. In this section, we assess the significance of this background information, as illustrated in Table 4.

### A.3   LIMITATION

The proposed DVDet has some potential limitations. Firstly, it has yet to integrate transformer-based methods. Secondly, while its approach is straightforward, it consists of some constant parameters. For the transformer-related limitation, we leave it for future research. Regarding the constant parameters, DVDet shares them when combined with other detectors. We believe that innovative techniques, such as uncertainty-aware mechanisms for selecting visually related descriptors, offer promising methods for addressing these issues and are worth deeper exploration.

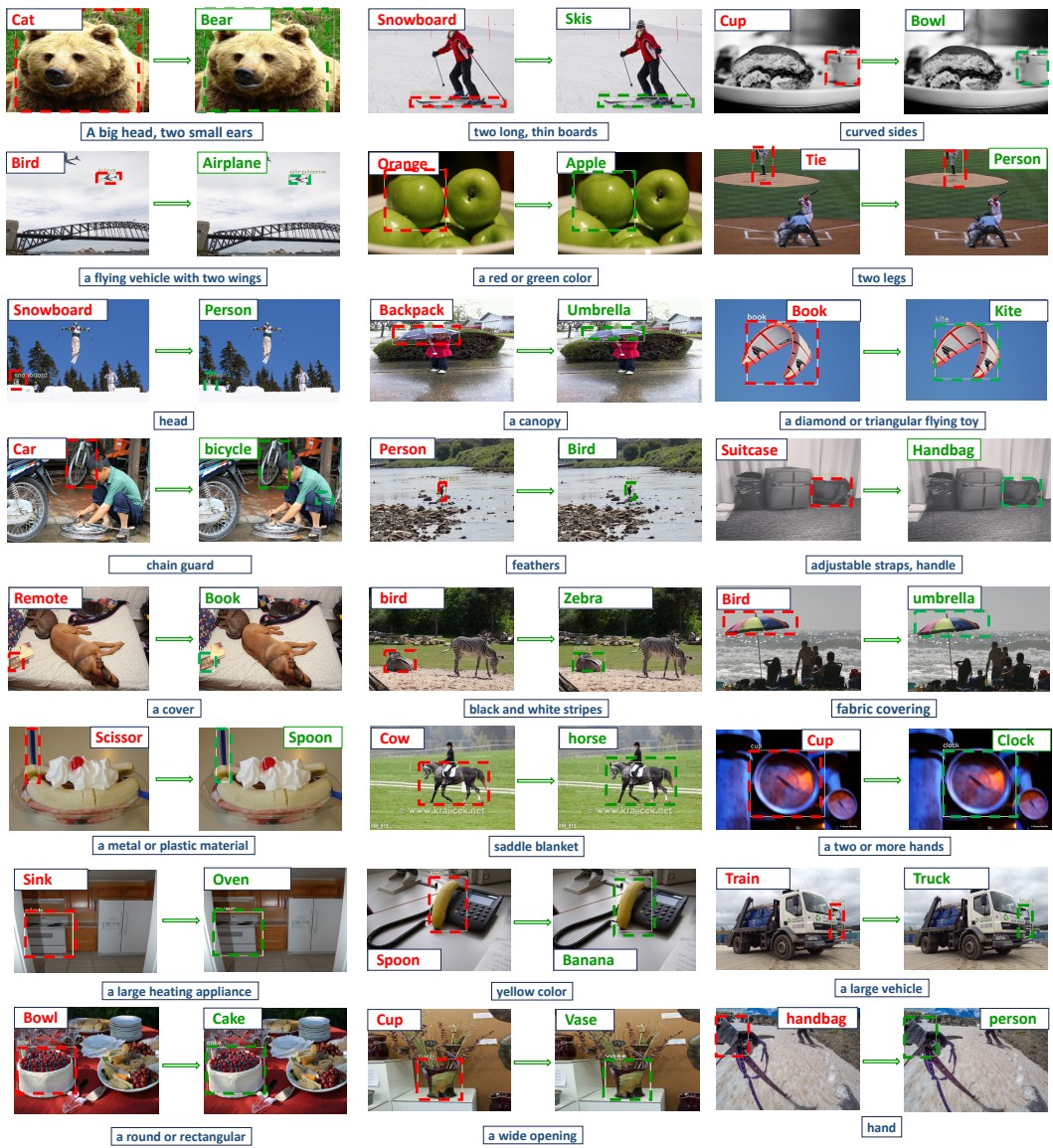

Figure 2: Introducing our fine-grained text descriptors (shown at the bottom of each sample) improves the open-vocabulary detection consistently under various scenarios. For each of these images, the red-color class names of the first image are predictions without our method, and the green-color class names in the second image are predictions after including our method. The red/green boxes in the sample images show related detection. Close-up view for details.

## A.4 VISUALIZATION RESULTS

In this section, we present extended visualization results for the proposed *DVDet*. Firstly, Fig. 2 demonstrates the enhancement brought by fine-grained text descriptors in open-vocabulary detection. Secondly, Fig. 3 offers deeper insights into the proficiency of our method in rectifying misclassifications. Furthermore, Fig. 4 categorizes and visualizes three specific types of failure cases: 1) the learned fine-grained alignment potentially introduces unintended consequences; 2) our method's learned prompt failed to address misclassifications; 3) the ground truth label presents ambiguity, leading to multiple interpretations.

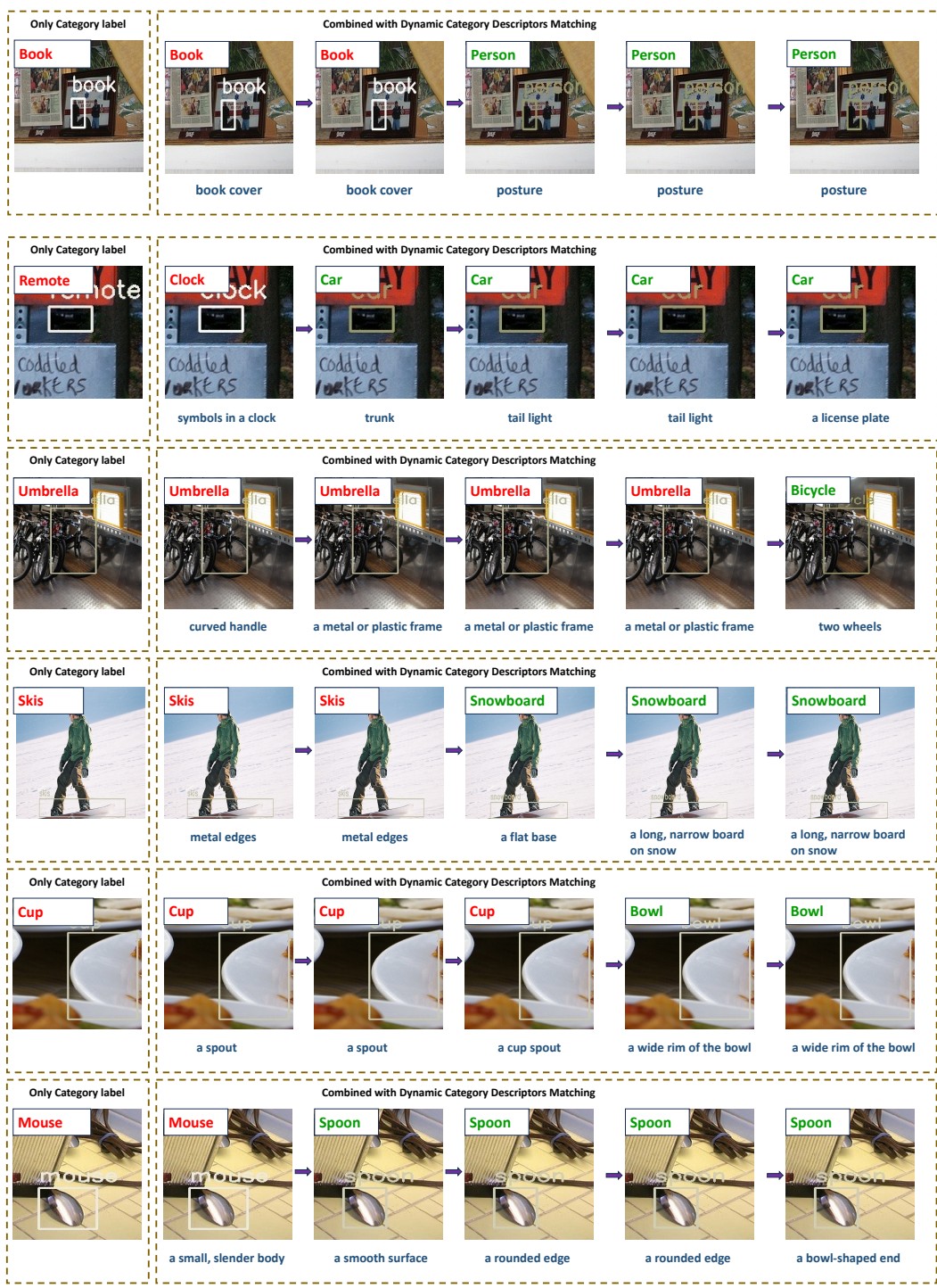

Figure 3: Object detection is improved progressively with iterative extraction of fine-grained descriptors from LLMs and matching them with the detected target. Texts at the top-left corner of each sample show recognized classes, and texts at the bottom show extracted fine-grained descriptors.

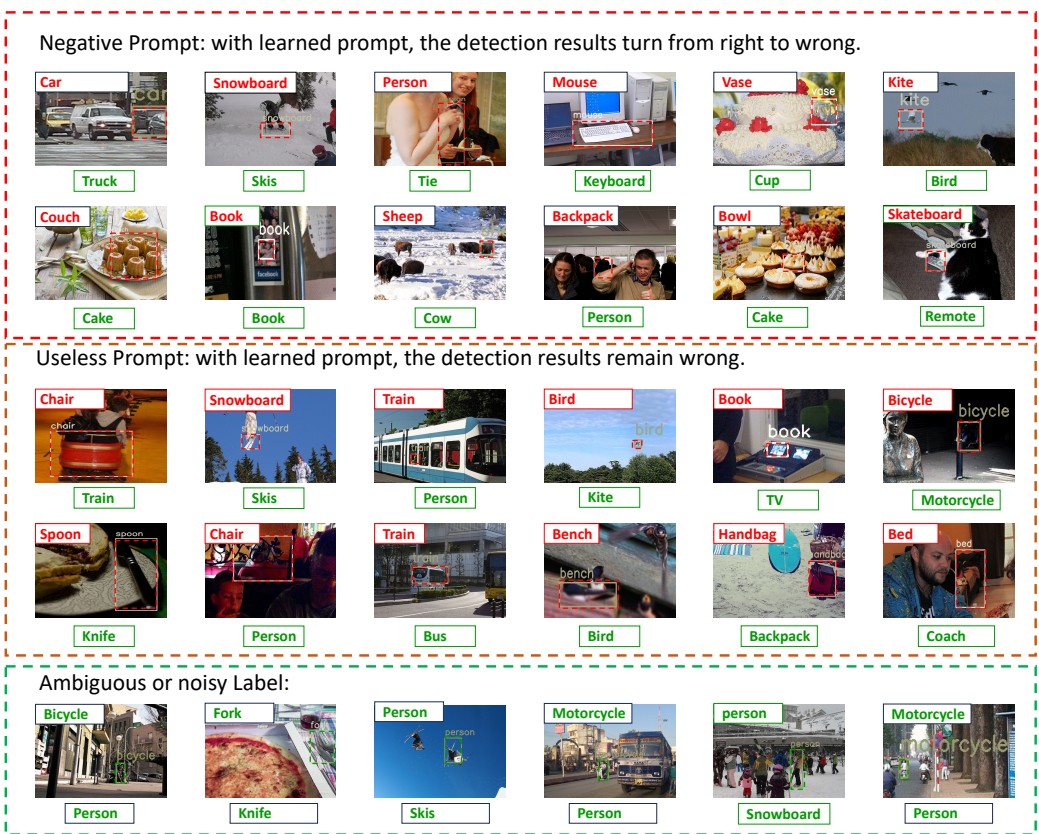

Figure 4: In this setting, we show visualization results of three types of failure cases. For each of these images, texts at the top-left corner of each sample show recognized classes, and texts at the bottom show the ground truth label. For each of the sample images, the red-color class names at the top-left corner of the image are misclassifications, and the green-color class names are right predictions. The red/green boxes within the sample images show related detection. Close-up view for details.