# OpenReview forum: "LLMs Meet VLMs: Boost Open Vocabulary Object Detection with Fine-grained Descriptors"
_ICLR.cc/2024/Conference — ICLR 2024 poster_

### Official Review · Reviewer_E6zT · 2023-10-31

**Soundness:** 3 good
**Presentation:** 3 good
**Contribution:** 3 good
**Rating:** 6
**Confidence:** 4

**Summary:**

This paper proposes Descriptor-Enhanced Open-Vocabulary Detector (DVDet) which introduces a conditional visual (region) prompting and textual descriptor dictionaries by using Large Language Models. The use of textual descriptors is motivated by the observation that the VLMs are good at capturing the fine-grained attributes of the objects. The textual descriptor dictionary is dynamically updated during training by tracking how frequent each descriptors are used, and by prompting what visual descriptors can help distinguishing the confusing categories. In addition, this work proposes a conditional region prompt to help the alignment between the detected region and text embeddings. When using both the textual descriptors and conditional visual prompts, DVDet improves the modern open-vocabulary detectors on the OV-COCO and OV-LVIS benchmarks.

**Strengths:**

* The use of Large Language Models to enrich the vocabulary description is well motivated.

* The proposed conditional visual prompt is useful in bridging the gap between the image-text and region-text alignment.

* The proposed DVDet method can be plugged into other SOTA open-vocabulary detection methods and consistently improve the performance.

**Weaknesses:**

* The use of region-level prompting for open-vocabulary detection is previously proposed in CVPR 2023 paper "CORA: Adapting CLIP for Open-Vocabulary Detection with Region Prompting and Anchor Pre-Matching". The comparison (both at a idea level and a numeric comparison) with CORA work seems necessary regarding the technical similarity.

* It is not clear what the optimal number of iterations is for the dictionary update. Is "the number of high-frequency descriptors" in Table 7 the number of iterations?

* What is the effect of using the LLM on the running speed of the DVDet detector (using no LLM vs N-th interaction of LLM dictionary)? While the running speed might not be the main focus of open-vocabulary detection research, the multiple iterations of LLM use could make the speed not very favorable.

* There are non-negligible number of hyper parameters (number of descriptors, number of update iterations, thresholds).

**Questions:**

* The gain in the base vs novel categories are similar (e.g., + 2.7 novel AP and +2.2 base AP in Table 1). This leads to a question whether the DVDet method is beneficial in standard detection instead of open-vocabulary detection. That is, the proposed method benefits from augmenting the vocabulary description (text embeddings) in general, not being specifically helpful for the novel categories. What is the motivation of evaluating on open-vocabulary detection? How would DVDet perform in the standard detection setting?

* Please compare with CORA (CVPR 2023) which also uses visual region prompting for open-vocabulary detection.

* Please refer to the weakness section.

---

> ### Author Response · Authors · 2023-11-19
> **Response to reviewer E6zT (Part I)**
>
> We are heartened by the reviewer's recognition of our method's motivation and utility. We value your insightful summary and constructive feedback and address each point as follows:
>
> > Q1: Comparison with region-level prompting in CORA (CVPR 2023)
>
> In contrast to CORA's approach, our visual prompts integrate background information from images. Table 9 of our paper shows that incorporating background information leads to performance improvements, illustrating the effectiveness of our method. We believe that this design offers a more comprehensive and contextual understanding of the image, aligning more closely with the visual prompt concept. Here are the experimental results:
>
> |  CCP  |   mAP_novel  |   mAP_base  |
> |:--------:|:---------:|:---------:|
> |    w/o Background Information               	| 33.4      	| 51.8  |
> | DVDet          	| **34.6**      	| **52.8**  |
>
> This distinction in prompt design underlines the originality of our approach and its effectiveness in enhancing detection performance.
>
> > Q2: Lack of Parameters Experiment on Update Frequency for Dictionary Update
>
> We would clarify that the update frequency of the descriptors dictionary is not equal to "the number of high-frequency descriptors". We conduct new experiments to assess its impact on performance. As the table below shows, DVDet maintains stable performance when the dictionary update frequency remains within a certain range
>
> |  Update Frequency of Dictionary |   0  |   100  |   200  |   400  |
> |:--------:|:---------:|:---------:|:---------:|:---------:|
> |    mAP_novel               	| 33.2      	| **34.8**  | 34.6      	| 34.6  |
> | mAP_base          	| 51.9     	| **53.2**  | 52.8      	| 52.6  |
>
> > Q3: Effect of LLM on DVDet's Running Speed
>
> * We appreciate the reviewer pointing out the need for experimental analysis on efficiency. In our study, the primary factor impacting training efficiency is the update of category descriptors. Notably, the update time for the descriptor dictionary accounts for only **20.16%** of the total training time when the update frequency is set at every 100 iterations in our implementation. This ratio demonstrates that our approach achieves an efficient balance, being comprehensive while remaining resource-effective.
>
> * We further conducted experiments to evaluate this, as shown in the table below. The results indicate that within a certain range, reducing the update frequency of the dictionary allows DVDet to maintain stable performance.
>
> |  Update Frequency of Dictionary |   0  |   100  |   200  |   400  |
> |:--------:|:---------:|:---------:|:---------:|:---------:|
> |    mAP_novel               	| 33.2      	| **34.8**  | 34.6      	| 34.6  |
> | mAP_base          	| 51.9     	| **53.2**  | 52.8      	| 52.6  |
>
> > Q4: Handling Multiple Hyperparameters
>
> Actually, our method focuses on three key parameters: the length 'm' of the enlarged proposals, the update frequency 'N' of descriptors, and the number 'TopN' of high-frequency descriptors. However, our experimental data indicate that the algorithm's performance is not highly sensitive to these parameters. This is supported by the results shown in Table 6 of our paper regarding the 'TopN' parameter. In response to Q2, we have analyzed the impact of the Descriptor Update Frequency 'N'. Additionally, to further demonstrate the importance of the parameter 'm', which determines the width and height of the proposal, we present the following experimental results:
>
> |  the values m and n  |   0  |   20  |   30  |   40  |
> |:--------:|:---------:|:---------:|:---------:|:---------:|
> |    mAP_novel     | 33.4    | 34.6  | 34.8  | 34.4  |
> |  mAP_base  |  51.8      	|  52.8  |  52.9  | 52.4  |
>
> Moving forward, we plan to implement learnable parameters and structures, aiming to reduce the necessity for manual hyperparameter tuning.

---

> > ### Author Response · Authors · 2023-11-19
> > **Response to reviewer E6zT (Part II)**
> >
> > > W1: Gains in Standard Detection vs. Open-Vocabulary Detection
> >
> > Thank you for highlighting this important aspect. While our method does not show more pronounced improvements in novel categories compared to base categories, it achieves consistent enhancements in both. As depicted in Figure 1 of our paper, unlike VLM-based classification models [1], current open-vocabulary detectors have difficulty aligning visual features with category descriptions. Our ablation studies reveal that descriptors do not boost the pretrained model during inference in detection tasks as effectively as in classification tasks, sometimes even leading to performance decline. Thus, we focus on aligning both base categories and descriptors during training. This approach yields improvements in novel categories, akin to the zero-shot performance seen in classification models, affirming our method's ability in enhancing open-set detection. As shown in our paper's Figure 5 and Figure 8, extensive visualizations further illustrate how descriptors progressively refine detection results in new categories.
> >
> > [1] Menon S, Vondrick C. Visual classification via description from large language models[J]. ICLR2023.

---

### Official Review · Reviewer_6yi2 · 2023-11-01

**Soundness:** 4 excellent
**Presentation:** 3 good
**Contribution:** 3 good
**Rating:** 6
**Confidence:** 4

**Summary:**

This paper introduces DVDet, a Descriptor-Enhanced Open Vocabulary Detector featuring two innovative components: conditional context prompts and hierarchical textual descriptors. The conditional context prompts function in the Region of Interest (RoI) align to expand each region proposal's area, incorporating more context information and converting it into an image-like feature preferred by CLIP. Meanwhile, the hierarchical textual descriptors dynamically acquire fine-grained descriptors for each category through interaction with the Language Model (LLM) during training. Extensive experiments indicate that DVDet significantly enhances the performance of various Open-Vocabulary Detectors on two benchmarks, MS COCO and LVIS.

**Strengths:**

1. I greatly value the use of LLM to dynamically generate detailed descriptors of categories. This approach inspires a novel method of employing LLM to enhance the performance of OVD tasks.

2. The concept presented is both straightforward and potent, with the paper being lucid and easy to navigate. The author's introduction of Framework Figures (Fig 2) and the examples of iterative fine-grained descriptors (Figure 3) are commendable as they significantly aid the reader in grasping the idea more effectively.

3. The experiments conducted are thoroughly robust. First, the author meticulously scrutinizes each component of their methodology, as detailed in Table 3 and Table 9. Second, the author conducts comprehensive experiments on a variety of OVD detectors and datasets to demonstrate that DVDet significantly enhances performance.

**Weaknesses:**

1. A comparison of computation costs with the baseline has not been conducted.

2. The contribution of the technique is somewhat restricted.

3. The prompt generated by LLM may possess an element of randomness, potentially making it challenging to reproduce.

4. No ablation experiment about the hyper-parameter $m$ of Conditional Context Prompt

**Questions:**

1. How about the extra computation overhead?

2. LLM may generate different prompts. Does it significantly affect the performance?

3. How about different $m$ in Conditional Context Prompt?

---

> ### Author Response · Authors · 2023-11-19
> **Response to reviewer 6yi2**
>
> We are grateful that the reviewer recognizes the value, straightforwardness, and effectiveness of our method. We appreciate your concerns about the algorithm's efficiency increase and stability and will address each of your points in turn.
>
> > Q1: A comparison of computation costs with the baseline.
>
> * We appreciate the reviewer pointing out the need for experimental analysis on efficiency. In our study, the primary factor impacting training efficiency is the update of category descriptors. Notably, the update time for the descriptor dictionary accounts for only **20.16%** of the total training time when the update frequency is set at every 100 iterations in our implementation. This ratio demonstrates that our approach achieves an efficient balance, being comprehensive while remaining resource-effective.
>
>
> * We further conducted experiments to evaluate this, as shown in the table below. The results indicate that within a certain range, reducing the update frequency of the dictionary allows DVDet to maintain stable performance.
>
> |  Update Frequency of Dictionary |   0  |   100  |   200  |   400  |
> |:--------:|:---------:|:---------:|:---------:|:---------:|
> |    mAP_novel               	| 33.2      	| **34.8**  | 34.6      	| 34.6  |
> | mAP_base          	| 51.9     	| **53.2**  | 52.8      	| 52.6  |
>
> > Q2: Prompt Generation Randomness from LLMs
>
> We are pleased that the reviewer has pointed out the impact of descriptor randomness on experimental performance. In fact, DVDet's interactive mechanism was designed with the generative model's randomness in mind. We have implemented two relevant mechanisms:
>
> * Descriptor Update Mechanism: By retaining high-frequency descriptors and discarding low-frequency ones, we ensure the visual relevance of the descriptors.
> * Descriptor Selection Mechanism: During the inference phase, we discard irrelevant descriptors, thereby enhancing stability.
> * We further present the performance variability across five experimental runs to demonstrate the robustness of our algorithm.
>
> |  LVIS  |   mAP_novel  |   mAP_all  |
> |:--------:|:---------:|:---------:|
> |    VLDet      | 27.5  $\pm$ 0.2      	| 41.8 $\pm$ 0.3 |
>
> > Q3: Lack of Ablation Experiment on Hyper-Parameter m of Conditional Context Prompt:
>
> * Firstly, in Table 9 of our paper, we provided ablation studies regarding background information, showing the experimental outcomes when parameter m is set to 0.
>
> * We conducted further experiments about the parameter m. As the table below shows, DVDet's performance remains stable when m lies within a certain range:
>
> |  the values m and n  |   0  |   20  |   30  |   40  |
> |:--------:|:---------:|:---------:|:---------:|:---------:|
> |    mAP_novel     | 33.4    | 34.6  | 34.8  | 34.4  |
> |  mAP_base  |  51.8      	|  52.8  |  52.9  | 52.4  |

---

### Official Review · Reviewer_shSW · 2023-11-01

**Soundness:** 3 good
**Presentation:** 3 good
**Contribution:** 2 fair
**Rating:** 6
**Confidence:** 5

**Summary:**

This paper proposes an approach to boost open-vocabulary object detector with fine-grained language descriptors of the categories. The motivation is that existing works learn to align region embeddings with category labels only, disregarding the capability of VLM to align with object parts and other fine-grained descriptions. The fine-grained text descriptors are generated with an LLM in an interactive manner. In addition, the paper introduces conditional context prompt to augment region embeddings with contextual cues and make them more image-like for open-vocabulary detection.

**Strengths:**

The two approaches presented by this paper are intuitive and effective. The context prompt makes the region more image-like to aid open-vocabulary recognition and the use of LLM provides additional features for VLM to reliably detect novel classes. The main results in Table 1 and 2 show the effectiveness of proposed approaches on LVIS and COCO benchmarks based on many existing methods. Ablations show that each part is effective on OV-COCO.

**Weaknesses:**

1. The idea of using LLM to generate more textual description has been explored in recent/concurrent work [2]. It'd be great to see a comparison with the existing ideas. Similarly, the idea of context regional prompt has been explored in recent/concurrent work [1], although the exact instantiation may differ. It'd be helpful to tease apart the contribution of context (which makes the crop more image-like) vs prompting (which adapts the features for detection use cases) through an ablation, and compare/discuss the similarity/differences with existing work. I understand the references listed are very recent/concurrent works, but I think it'd still be valuable to have some comparison with them for scientific understanding.

2. Although DVDet shows gains for all methods in Table 1 and 2. The gains on base categories are non-trivial and even comparable to the novel categories in most cases. For example, DVDet + Detic has +1.8 boost on mAPf and +1.3 on mAPr, and +2.5 on Base AP vs +1.7 on Novel AP. Given the method is based on Detic, I'm wondering what the implication of the gains on base vs novel categories are, since the method is primarily designed for open-vocabulary detection.

3. Table 3 (row 2) shows that adding fine-grained descriptors by itself without prompting hurts the performance of the model. This seems counter-intuitive to me. It'd be helpful to see if the same observation holds for other OVD models e.g. ViLD or Detic.

4. Table 4 shows transfer detection from COCO to PASCAL and LVIS. It's nice to see a clear boost there. I'd recommend trying out the LVIS-trained model on Objects365 instead since it's a more commonly used transfer detection setting by e.g. ViLD, DetPro, F-VLM.

5. In Eq (2), the $\textit{m, n}$ are said to be constants. How are they set? I'm wondering if they should be set proportional to the width and height of the ROI. Another option is to use the whole image box as a baseline and see how that performs. Some ablations/analysis on the choice of context would be interesting.

References:
1. CORA: Adapting CLIP for Open-Vocabulary Detection with Region Prompting and Anchor Pre-Matching (CVPR 2023)
2. Multi-Modal Classifiers for Open-Vocabulary Object Detection (ICML 2023)

**Questions:**

See weaknesses. Point 1-3 are more important in my view.

What's the variance of the proposed method on LVIS open-vocabulary benchmark over e.g. 3 or 5 independent runs?

---

> ### Author Response · Authors · 2023-11-19
> **Response to reviewer shSW (Part. I)**
>
> We are really encouraged that the reviewer recognizes our method to be intuitive and effective.
>
> We thank the valuable comments and insightful suggestions, and we hope our detailed responses below can address your concerns.
>
> > W1: Discussion with Concurrent Work [1] and [2].
>
> As the reviewer noted, our work mainly involves the generation of category descriptors and the regional visual prompt. We appreciate the reviewer pointing out the need to compare our approach with the latest methods CORA [1] and MCVOD [2], and we provide a comparison with each below.
>
> Firstly, regarding the generation of descriptors, our approach differs significantly from MCVOD [2] in the way of interacting with LLMs. Specially, we leverage the interactive capabilities of LLMs to dynamically generate visually relevant descriptors for each category, rather than treating the process as a one-time interaction. As shown in Table 5 of our paper, introducing this interaction mechanism generally leads to a performance gain of around 1.5%. The experimental results are as follows:
>
> |  Interaction Strategy  |   mAP_novel  |   mAP_base  |
> |:--------:|:---------:|:---------:|
> |    Static               	| 33.2      	| 51.9  |
> | Interactive (DVDet)          	| **34.6**      	| **52.8**  |
>
> Secondly, regarding visual prompts, our design differs from the prompts in CORA [1] by integrating background information from images. As Table 9 of our paper shows, we observe further performance improvements when the background information is incorporated. As the reviewer pointed out, prompts without background information can be considered tailored for detection tasks (**prompting**). Incorporating background information further merges contextual information (**context**), making it more consistent with the image. The experimental results are as follows:
>
> |  CCP  |   mAP_novel  |   mAP_base  |
> |:--------:|:---------:|:---------:|
> |    w/o Background Information               	| 33.4      	| 51.8  |
> | DVDet          	| **34.6**      	| **52.8**  |
>
> We will add the discussion with the concurrent methods[1, 2] in our updated manuscript.
>
> [1] Wu X, Zhu F, Zhao R, et al. CORA: Adapting CLIP for Open-Vocabulary Detection with Region Prompting and Anchor Pre-Matching[C]. CVPR 2023.
>
> [2] Kaul P, Xie W, Zisserman A. Multi-Modal Classifiers for Open-Vocabulary Object Detection [C]. ICML 2023
>
>
> > W2: Gains on Base vs. Novel Categories.
>
> Thank you for highlighting this important point. We acknowledge that our method does not show much more improvements in novel categories compared to base categories. Nonetheless, it's notable that our algorithm achieves consistent enhancements in both.
>
> As illustrated in Figure 1 of our paper, unlike classification models using VLMs, current open-vocabulary detectors struggle to align visual features with descriptions of categories. Our ablation studies in Table3 of our paper, visualized in the results, demonstrate that category descriptors do not boost the pretrained model in detection tasks as effectively as they do in classification tasks [3], occasionally leading to reduced performance.
>
> In our training process, we focus on aligning both base categories and descriptors, endowing them with better cross-modal alignment capability. This approach further yields improvements in novel categories, thereby confirming our method's effectiveness in enhancing open-set detection performance. The visualizations in Figure 5 of our paper further demonstrate how descriptors progressively refine detection results in new categories.
>
> [3] Menon S, Vondrick C. Visual classification via description from large language models[J]. ICLR2023.
>
> > W3: Decreased Performance with Descriptors without Prompt Learning
>
> Thank you very much for the constructive suggestion. The reason for this phenomenon, where adding fine-grained descriptors by itself without prompting hurts the performance of the model, lies in the disparity between the capabilities of VLMs and current open vocabulary detectors in cross-modal alignment. As shown in Figure 1 of our paper, the visual features extracted by VLMs maintain high consistency with the textual embeddings of descriptors. Therefore, in classification tasks [3], they can directly enhance performance in the inference stage without additional training. However, this is challenging to achieve in detection tasks. To make this conclusion more convincing, we have also compiled results on Detic and provided the following ablation study:
>
> |  Detic  |   mAP_novel  |   mAP_base  |
> |:--------:|:---------:|:---------:|
> |    w/o Prompt Learning               	| 25.5      	| 47.9  |
> | Prompt Learning          	| **27.8**      	| **51.1**  |
>
> We will update the experiments and discussions.

---

> > ### Author Response · Authors · 2023-11-19
> > **Response to reviewer shSW (Part. II)**
> >
> > > W4: Transfer LVIS Detection Model to Objects365.
> >
> > Thank you for your suggestion. We conducted the suggested experiments and the experimental results are presented below, where we conduct the experiments on representative DetPro as suggested:
> >
> > |  Objects365  |   AP_s  |   AP_m  |   AP_l  |
> > |:--------:|:---------:|:---------:|:---------:|
> > |    DetPro               	| 4.5      	| 11.5  | 18.6  |
> > | +DVDet         	| **5.1**      	| **12.3**  | **19.8**  |
> >
> > > W5: Setting Constants m and n in Eq (2):
> >
> > * In our paper, the values of m and n are both fixed at 20. We present experimental results about the parameters m and n, illustrating that DVDet's performance remains stable when the two parameters lie within a certain range:
> >
> > |  the values m and n  |   0  |   20  |   30  |   40  |
> > |:--------:|:---------:|:---------:|:---------:|:---------:|
> > |    mAP_novel     | 33.4    | 34.6  | 34.8  | 34.4  |
> > |  mAP_base  |  51.8      	|  52.8  |  52.9  | 52.4  |
> >
> > > Q: In response to the query about variance on the LVIS Open-Vocabulary Benchmark, we conducted three independent runs to assess the consistency of our method. The experiments show that the variance is very small, indicating the stability of our approach:
> >
> > |  LVIS  |   mAP_novel  |   mAP_all  |
> > |:--------:|:---------:|:---------:|
> > |    VLDet      | 27.5  $\pm$ 0.2      	| 41.8 $\pm$ 0.3 |

---

### Official Review · Reviewer_AtsG · 2023-11-02

**Soundness:** 3 good
**Presentation:** 3 good
**Contribution:** 3 good
**Rating:** 6
**Confidence:** 4

**Summary:**

This paper proposes to a novel open vocabulary detection method which uses a fine-grained descriptor for a better region-text alignment and open vocabulary detection. It consists of two parts. First part is to transform region embeddings to image-like representation, second part is to use LLM  to generate fine-grained descriptors. The method improves open vocabulary detection consistently.

**Strengths:**

1. The method proposed in the paper looks interesting by using LLM as a better fine-grained descriptor. The idea is reasonable and the presentation is good.
2. The progress from iterative extraction of fine-grained LLM descriptor is interesting and looks interesting to me.
3. The results of experiments are good and the ablation study is convincing to me.

**Weaknesses:**

1. when LLM interacts with VLM, Could you give more examples of prompts to use and how the prompt may influence the final results?
2. when iterative updating of LLM descriptor, It seems to just give a general description of objects. Then what will happen if we just let LLM do some general description of objects without seeing the objects?
3. if the regional prompt is not accurate, how will the performance look like when interacting with LLM?

**Questions:**

please see the weakness.

---

> ### Author Response · Authors · 2023-11-19
> **Response to reviewer AtsG**
>
> We are grateful for the reviewer's interest in understanding how descriptors and regional prompts influence detection results. We address each point of interest in turn.
>
> > W1: when LLM interacts with VLM, Could you give more examples of prompts to use and how the prompt may influence the final results:
>
> In this paper, we utilize three prompts to interact with LLMs to gather descriptors, 'Template N' only includes the category name, ‘Template H’ [1] includes high-frequency descriptors and the category name, and ‘Template C’ includes confusing categories and the category name. The performance using different prompts is as follows:
>
> |  Interaction Strategy  |   mAP_novel  |   mAP_base  |
> |:--------:|:---------:|:---------:|
> |    VLDet               	| 32.0      	| 50.6  |
> |    +template N               	| 33.1      	| 51.2  |
> |    +template C               	| 33.8      	| **52.3**  |
> |    +template H               	| **34.0**      	| 52.2  |
>
> Our experimental findings show that our designed templates, namely templates C and H, are more effective than the template N used in [1] for prompting LLMs. This results in the generation of effective descriptors and an overall improvement in performance, highlighting the efficacy of our method in descriptor generation.
>
> [1] Menon S, Vondrick C. Visual classification via description from large language models[J]. ICLR2023.
>
>
> > W2: What will happen if we just let LLM do some general description of objects without seeing the objects?:
>
> * Firstly, in our method, the Large Language Models (LLMs) understand objects through their interaction with Vision Language Models (VLMs). So, during training, the LLMs don't see novel classes. The improvement in detecting these novel classes, which are unknown to LLMs, shows how effective our generated descriptors are.
>
> * Secondly, we present experimental results where all classes were unknown to the LLMs. The results from Table 5 of our paper are as follows:
>
> |  Interaction Strategy  |   mAP_novel  |   mAP_base  |
> |:--------:|:---------:|:---------:|
> |    VLDet               	| 32.0      	| 50.6  |
> |    Static               	| **33.2**      	| **51.9**  |
>
>
> These results indicate that even without interacting with detection models, the descriptors from LLMs still improve the performance of these models.
>
> > W3: if the regional prompt is not accurate, how will the performance look like when interacting with LLM?
>
> As the reviewer correctly points out, the accuracy of prompts can indeed affect the algorithm's performance. As shown in Table 3 of our paper, we present the performance of DVDet with Prompt Learning. To further illustrate the impact of prompt accuracy, we added a new experiment using a Random Prompt. This additional experiment clearly demonstrates that inaccurate, randomly selected prompts lead to decreased performance. In contrast, our original approach with prompt learning significantly improves detection accuracy.
>
> |  Regional Prompt  |   mAP_novel  |   mAP_base  |
> |:--------:|:---------:|:---------:|
> |    VLDet               	| 32.0      	| 50.6  |
> |    +random prompt | 27.3      	| 45.2  |
> |    +prompt learning | **33.1**      	| **51.2**  |

---

### Official Review · Reviewer_qAZ8 · 2023-11-07

**Soundness:** 3 good
**Presentation:** 3 good
**Contribution:** 3 good
**Rating:** 6
**Confidence:** 4

**Summary:**

The submission introduces Descriptor-Enhanced Open Vocabulary Detection (DVDet), a method that integrates fine-grained descriptors from Vision Language Models (VLMs) into open vocabulary object detection (OVOD) using a Conditional Context visual Prompt (CCP). This approach utilizes large language models to generate descriptors without extra annotations and features a hierarchical update mechanism for descriptor refinement. The paper claims improvements in OVOD tasks through extensive experiments and presents three main contributions: a new feature-level visual prompt, an update mechanism for descriptor management, and empirical evidence of enhanced detection performance.

**Strengths:**

+ Introduction of Descriptor-Enhanced Open Vocabulary Detection (DVDet) could address existing granularity challenges in object detection, making it a potentially transformative method.
+ Leveraging large language models for descriptor generation without additional annotations presents a cost-effective solution.

**Weaknesses:**

- The paper does not sufficiently address the analysis of descriptors produced by Large Language Models, which is crucial for understanding the quality and relevance of the generated descriptors in enhancing object detection.
- The demonstrated enhancements (~2% on average) in object detection performance, while positive, do not represent a substantial leap forward when considering the benchmarks established by current leading methods.

**Questions:**

1. How does the descriptors generation process affects the training efficiency?
2. Can the authors provide insights into any observed trade-offs between the complexity of the Descriptor-Enhanced Open Vocabulary Detection (DVDet) and its performance gains?
2. How does the system perform under varying conditions, such as different object scales, occlusions, and lighting? Are there specific scenarios where the performance improvement is more pronounced?

---

> ### Author Response · Authors · 2023-11-19
> **Response to reviewer qAZ8**
>
> We thank the valuable comments and insightful suggestions, and we hope our detailed responses below can address your concerns.
>
> > W1: Analysis of Descriptors from LLMs, essential for evaluating their quality and impact on improving object detection.
>
> Thank you for highlighting the importance of descriptor analysis, which is indeed a central aspect of our work. Effectively generating descriptors through LLMs forms the core of our approach.
> *  Firstly, we designed an interactive mechanism for refining descriptors through ongoing interactions with LLMs. In contrast to static descriptors, which are produced through a one-time interaction, our interactive approach updates existing descriptors based on their contributions in enhancing detection performance. This point is well demonstrated in the experiment table below:
>
> |  Interaction Strategy  |   mAP_novel  |   mAP_base  |
> |:--------:|:---------:|:---------:|
> |    Static               	| 33.2      	| 51.9  |
> | Interactive          	| **34.6**      	| **52.8**  |
>
> * Secondly, our semantic selection mechanism chooses visual-related descriptors, thereby avoiding the influence of descriptors that are not visible in the current sample. This design highlights the importance of the relevance of the descriptors as validated in the table below.
>
> |  TopN  |   0  |   1  |    3  |   5  |  10  |   15  |
> |:--------:|:---------:|:---------:|:---------:|:---------:|:---------:|:---------:|
> |    mAP_base               	| 50.2      	| 51.2  | 52.0    | **52.8**  |51.9      	| 51.7  |
> | mAP_novel          	| 32.0      	| 32.4  |34.4      	| **34.6**  |33.0      	| 32.2  |
>
> * Lastly, we provide extensive visualization in Figures 7 and 8 of our paper, which illustrate how the descriptors improve detection performance. These visualizations clearly demonstrate the effectiveness of our descriptor generation in enhancing object detection performance.
>
>
> > W2: The performance enhancements (~2% on average) in object detection.
>
> We would like to share that a 2% improvement in performance metrics is quite acceptable in the current setting. For instance, methods like VLDet [1] which incorporates caption data also achieve only around 2% improvement compared to those without caption data. However, we would like to reiterate two key points: first, our method employs lightweight prompt learning, and second, it achieves consistent improvements across multiple detectors.
>
> [1] Lin C, Sun P, Jiang Y, et al. Learning object-language alignments for open-vocabulary object detection[J]. ICLR, 2023.
>
> > Q1: The training efficiency affected by the descriptors generation process.
>
> We appreciate the reviewer pointing out the need for experimental analysis on efficiency. In our study, the primary factor impacting training efficiency is the update of category descriptors. Notably, the update time for the descriptor dictionary accounts for only **20.16%** of the total training time when the update frequency is set at every 100 iterations in our implementation. This ratio demonstrates that our approach achieves an efficient balance, being comprehensive while remaining resource-effective.
>
> > Q2: Complexity and Performance Trade-offs:
>
> Building on our previous discussion, we identified that the frequency of dictionary updates (i.e., the frequency of interactions with LLMs per number of iterations) is a key factor influencing training efficiency. We conducted experiments to evaluate this, as shown in the table below. The results indicate that within a certain range, reducing the update frequency of the dictionary allows DVDet to maintain stable performance.
>
> |  Update Frequency of Dictionary |   0  |   100  |   200  |   400  |
> |:--------:|:---------:|:---------:|:---------:|:---------:|
> |    mAP_novel               	| 33.2      	| **34.8**  | 34.6      	| 34.6  |
> | mAP_base          	| 51.9     	| **53.2**  | 52.8      	| 52.6  |
>
> > Q3: Performance Under Varying Complex Conditions
>
> In Figure 7 of our paper, we demonstrate how fine-grained descriptors correct the model's detection results. Further visualizations of this process are provided in Figure 8 of our paper. These visualizations clearly show that our approach significantly improves detection, particularly in challenging scenarios. These scenarios include (1) detecting distant or occluded objects, and (2) distinguishing objects with small inter-class variations.

---

### Author Response · Authors · 2023-11-19
**Thanks for reviewers' valuable time and insightful suggestion**

We thank all the reviewers for their time, insightful suggestions, and valuable comments. We are encouraged to see that ALL reviewers recognize our method for its transformative and cost-effective (qAZ8), interesting nature and good presentation (AtsG), intuitiveness (shSW), straightforwardness and potency (6yi2), and well-motivated approach (E6zT). The majority of reviewers acknowledged the value of interacting with large-scale models.

However, there are additional concerns from the reviewers, mainly focusing on:

* More comparisons and discussions with the latest existing methods.
* Further analysis of experimental results.
* Additional experiments regarding efficiency and parameter settings.

We respond to each reviewer's comments in detail below. We again thank the reviewers for their valuable suggestions, which we believe will greatly strengthen our paper.

---

### Meta-Review · Area_Chair_BR3s · 2023-12-10

**Metareview:**

This paper was reviewed by five experts in the field. After authors' rebuttal, all reviews recommended acceptance. Reviewers liked the overall proposed method and strong performance gain.

The AC agrees with the reviewers' assessments and believes the paper provides an interesting method for enhancing VLM-based open-vocabulary detection with fine-grained descriptions of object categories. The reviewers did raise some valuable suggestions in the discussion. The AC believes the authors' responses greatly improve the paper and those new results and discussions should be incorporated into the final paper and supplementary material.

**Justification For Why Not Higher Score:**

The AC agrees with the reviewers (shSW, E6zT) that the related work CORA and MCVOD already cover part of the knowledge advancement introduced in this paper. This is the major reason why the paper doesn't merit a spotlight or oral presentation. In addition, the current form of the paper still needs improvement: various concerns have been raised by the reviewers and should be incorporated into the final paper.

**Justification For Why Not Lower Score:**

The reviewers unanimously recommended accepting the paper. The AC didn't find reasons to overturn the reviewers' consensus.

---

### Decision · Program_Chairs · 2024-01-16

Accept (poster)